**EMBO** *reports*

# NAD(P) transhydrogenase has vital non-mitochondrial functions in malaria parasite transmission

Sadia Saeed[1], Annie Z Tremp[1], Vikram Sharma[2], Edwin Lasonder[2] & Johannes T Dessens[1,*] (iD)

## Abstract

Nicotinamide adenine dinucleotide (NAD) and its phosphorylated form (NADP) are vital for cell function in all organisms and form cofactors to a host of enzymes in catabolic and anabolic processes. NAD(P) transhydrogenases (NTHs) catalyse hydride ion transfer between NAD(H) and NADP(H). Membrane-bound NTH isoforms reside in the cytoplasmic membrane of bacteria, and the inner membrane of mitochondria in metazoans, where they generate NADPH. Here, we show that malaria parasites encode a single membrane-bound NTH that localises to the crystalloid, an organelle required for sporozoite transmission from mosquitos to vertebrates. We demonstrate that NTH has an essential structural role in crystalloid biogenesis, whilst its enzymatic activity is required for sporozoite development. This pinpoints an essential function in sporogony to the activity of a single crystalloid protein. Its additional presence in the apicoplast of sporozoites identifies NTH as a likely supplier of NADPH for this organelle during liver infection. Our findings reveal that *Plasmodium* species have co-opted NTH to a variety of non-mitochondrial organelles to provide a critical source of NADPH reducing power.

**Keywords** malaria; oocyst; ookinete; sporozoite; transhydrogenase; transmission

**Subject Categories** Metabolism; Microbiology, Virology & Host Pathogen Interaction

## Introduction

Curbing malaria parasite transmission by mosquitoes is considered an essential part of successful malaria control and eradication programmes. In the parasite life cycle (Fig EV1), transmission starts with the uptake of haploid sexual stage precursor cells (gametocytes) from the vertebrate host with the blood meal of a feeding mosquito, setting off a rapid process of gametogenesis and fertilisation in the mosquito midgut. The resulting diploid zygotes undergo meiosis and transform into motile elongated forms called ookinetes, which traverse the midgut epithelium and then round up to form oocysts (Fig EV1). In the ensuing weeks, these young oocysts grow and divide by a process called sporogony generating thousands of haploid progeny cells named sporozoites. After egress from the oocysts, sporozoites colonise the insect's salivary glands, after which they are transmissible to new hosts by mosquito bite to infect liver cells and initiate new malaria blood-stage infections (Fig EV1).

NAD(P) transhydrogenases (NTHs) are enzymes that catalyse the reversible hydride ion transfer between NAD(H) and NADP(H). They exist as soluble (EC 1.6.1.1) and membrane-bound (EC 1.6.1.2) isoforms. The latter are integral multi-pass membrane proteins that in bacteria reside in the cytoplasmic membrane, whilst in metazoans they are situated in the inner membrane of mitochondria [1,2] likely reflecting the evolutionary origin of the mitochondrion from a bacterial primary endosymbiont [3,4]. In membrane-bound NTH proteins, the electron transfer reaction between NAD(H) and NADP(H) is coupled to the simultaneous translocation of a proton across the membrane in which it is embedded, following the reaction:

$$NADH + NADP^+ + H_{out}^+ \Leftrightarrow NAD^+ + NADPH + H_{in}^+$$

where 'in' and 'out' denote the cytosol and periplasmic space of bacteria, or the matrix and cytosol of the mitochondria, respectively [1,2,5]. The widely accepted view of the physiological role of mitochondrial NTH is to generate NADPH (the reduced form of NADP), either required for mitochondrion-specific biosynthetic purposes, or to protect the organelle from oxidative damage caused by free radicals generated in the respiratory chain [6,7]. In support of the latter, NTH-deficient *Caenorhabditis elegans* and NTH-deficient C57BL/6J mice show increased sensitivity to mitochondrial oxidative stress [8,9].

In this study, we characterise a membrane-bound NTH in malaria parasites that is not present in mitochondria, but instead localises in the crystalloid, an enigmatic organelle found in ookinetes and young oocysts that is critically involved in sporogony [10–13]. We show that NTH has an essential structural role in

1 Department of Infection Biology, Faculty of Infectious and Tropical Diseases, London School of Hygiene & Tropical Medicine, London, UK
2 School of Biomedical Sciences, University of Plymouth, Plymouth, UK
*Corresponding author. Tel: +442079272865; Fax: +442076374314; E-mail: johannes.dessens@lshtm.ac.uk

crystalloid formation, as well as a vital enzymatic role in sporogony, indicating that the organelle requires NADPH to function. NTH is also found in the sporozoite apicoplast, addressing a longstanding question about the potential source of NADPH required for some of the anabolic activities that take place in this plastid of likely red algal origin [14].

## Results and Discussion

### *Plasmodium* encodes a single, membrane-bound NTH

Genome analysis shows that *Plasmodium* species encode a single, conserved membrane-bound NTH (e.g. PlasmodDB identifiers PF3D7_1453500; PVX_117805; PBANKA_1317200). *Plasmodium berghei* NTH is encoded by a three-exon gene and is composed of 1,201 amino acids with a calculated $M_r$ of 135,198 and has a predicted amino-terminal ER signal peptide sequence that forms part of a bipartite apicoplast targeting sequence (PATS prediction: score 0.947 out of 1.000 [15]; PlasmoAP prediction: score 4 out of 5 tests positive [16]; Fig 1A). The gene product forms 11 predicted transmembrane helices and has additional domains for binding NAD(H) and NADP(H) (Fig 1A). This structure conforms with the general architecture of proton-translocating NTH proteins, consisting of three functional domains: domain I, which binds NAD(H); domain II, which contains the membrane-spanning helices, and domain III, which binds NADP(H) [2] (Fig 1A and B). Functional domains I and III together facilitate hydride transfer between NAD(H) and NADP(H), whereas domain II facilitates proton translocation across the lipid bilayer in which the NTH protein is embedded (Fig 1B). Whilst prokaryotes have segmented *nth* genes, eukaryotic *nth* genes are unsegmented and encode single polypeptide NTH proteins, either linking the α subunit C-terminus to the β subunit N-terminus (αβ) as illustrated by mammalian NTH or linking the β subunit C-terminus to the α subunit N-terminus (βα) as illustrated by *Plasmodium* NTH (Fig 1C). Accordingly, the order of domains in *Plasmodium* NTH is dIIa-dIII-dI-dIIb (Fig 1A).

### *Plasmodium* NTH is present in the crystalloid organelle

Various *Plasmodium* transcriptome studies identified transcripts of the *nth* gene predominantly in female gametocytes and to be translationally repressed [17–19]. This was a strong indication that the NTH protein is expressed in zygotes/ookinetes. To determine NTH protein expression and subcellular localisation in live *P. berghei* parasites, a genetically modified parasite was generated by double homologous crossover recombination that stably expresses, from its native promoter, the full length NTH fused at its carboxy-terminus to GFP (Fig EV2). The resulting parasites (termed NTH/GFP) developed normally in mouse and mosquito and were transmitted by mosquito bite, indicating that the GFP tag had not interfered with NTH function (assuming *nth* disruption gives a clear phenotype). In the mouse, neither asexual nor sexual blood-stage parasites displayed discernible GFP-based fluorescence consistent with the *nth* transcriptome data [17–19]. Dispersed extranuclear GFP fluorescence was first detected in zygotes ~ 4 h after gametocyte activation (Fig 2A) consistent with a post-fertilisation lifting of translational repression and a localisation of the protein in the ER. By 24 h,

mature ookinetes showed GFP fluorescence that was concentrated in typically two spots associated with pigment clusters (Fig 2B). This distribution pattern is characteristic of proteins that are trafficked to the crystalloids [11–13,20].

Previous studies have shown that a protein complex of six modular proteins rich in putative carbohydrate binding domains, named LCCL lectin adhesive-like proteins (LAPs), resides in the crystalloid [11,20,21]. To confirm the localisation of NTH in the crystalloid, we co-localised NTH with LAP3. To achieve this, LAP3 was fused to the red fluorescent protein mCherry [13] and the resulting parasite line (named LAP3/mCherry) was genetically crossed *in vitro* with parasite line NTH/GFP. Subsequent infection of mosquitoes with the resultant ookinete population gives rise to heterokaryotic polyploid oocysts containing mixtures of parental, wild-type, and double gene-tagged (i.e. possessing modified alleles for both *nth* and *lap3*) sporozoites (Fig 2C). The ensuing sporozoite population was transmitted to naive mice by mosquito bite and drug selected. Ookinete cultures derived from the transmitted parasite population contained zygotes and ookinetes with dual expression of NTH::GFP and LAP3::mCherry that co-localised both before and after crystalloid formation, respectively ($n = 50$; Fig 2D and E).

Western blot analysis of purified ookinetes of parasite line NTH/GFP with anti-GFP antibodies produced a single band of a size corresponding to the NTH::GFP fusion protein (~ 160 kDa; Fig 2F), confirming that the *nth* allele had been successfully tagged. Ookinetes that were *in vivo* crosslinked prior to Western blot analysis showed an additional high molecular weight band migrating at approximately double the molecular size, which possibly corresponds to the NTH homodimer. This is consistent with reports that NTH proteins in other organisms operate as dimers [5,22–25].

### *Plasmodium* NTH is required for sporogony

To study the function of NTH and its contribution to parasite development and infectivity, we generated a null mutant parasite line by double homologous crossover recombination in which the coding sequence of *nth* was removed (Fig EV2). The resultant parasites, named NTH-KO, developed normally in the mouse consistent with the absence of NTH expression in these life stages and with a recent report that this gene is dispensable for asexual blood-stage development in *P. berghei* [26]. NTH-KO parasites displayed normal gametocyte development, gametogenesis, and formed ookinetes of normal size and shape (Fig 3A). However, pigment in these ookinetes was more dispersed ($n = 50$) and not found in clusters that typically surround and highlight the crystalloids [11,27] (Fig 3A). Absence of these pigment clusters coincides with, and is caused by, loss of crystalloid formation as demonstrated recently for null mutants of the crystalloid protein DHHC10 [13]. In *Anopheles stephensi* mosquitoes, NTH-KO parasites developed oocyst numbers comparable to their wild-type counterparts (Figs 3B and EV3A). However, despite undergoing substantial nuclear expansion these oocysts failed to produce sporozoites (1,000 oocysts examined across at least 10 midguts from two independent infections; Fig 3C) and displayed increased oocyst size (Fig 3D and EV3B). Similar phenotypes have been reported for several LAP null mutants [11,12,28–30]. These collective findings demonstrate that NTH is required for crystalloid biogenesis and sporozoite development.

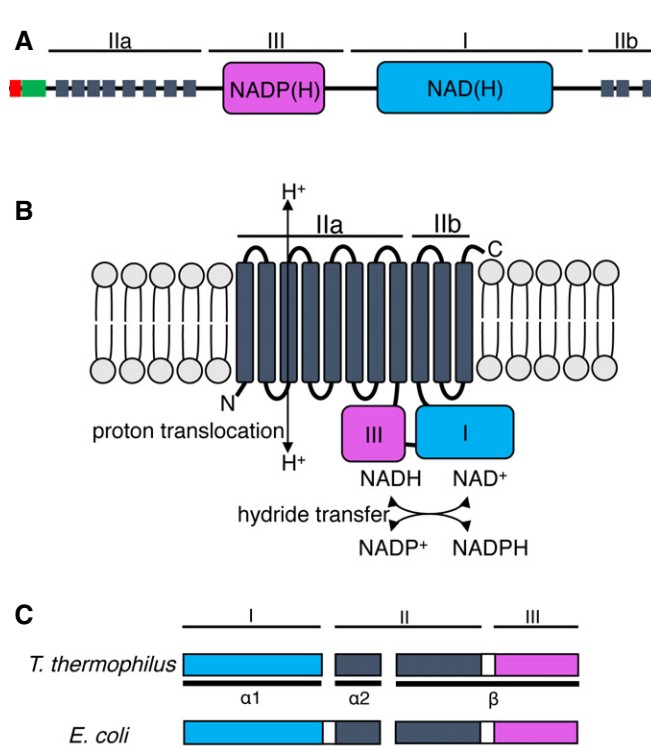

**Figure 1. Structure of the *Plasmodium nth* gene and gene product.**

A Schematic domain composition of PBANKA_1317200, showing predicted ER signal peptide (red), apicoplast transit peptide (green), transmembrane helices (dark grey), NADP(H) binding (pink), and NAD(H) binding (blue) modules. Functional NTH domains I, II, and III are indicated.

B Predicted functions of the *Plasmodium berghei* NTH protomer in the lipid bilayer showing functional domains I–III and corresponding functional activities.

C Organisation of functional NTH domains I–III (coloured bars) in *Thermus thermophilus*, *Escherichia coli*, mammals, and *Plasmodium*. Linker connecting functional domains is indicated by dotted line. Prokaryotic *nth* genes are segmented into three (*T. thermophilus*) or two (*E. coli*) segments, encoding NTH subunits α1, α2, and β (*T. thermophilus*) or α and β (*E. coli*). Eukaryotic *nth* genes are unsegmented and encode single polypeptide NTH of different orientations, either corresponding to an αβ linkage (mammal) or to a βα linkage (*Plasmodium*). The amino (N) and carboxy (C) termini of the eukaryotic NTH polypeptides are indicated.

A different parasite line named NTHΔPP was generated that expresses an N-terminally truncated version of NTH::GFP in which 60 amino acids were removed downstream of the ER signal peptide, including the predicted apicoplast transit peptide (Fig EV2). NTHΔPP ookinetes displayed weak dispersed GFP fluorescence that did not localise in crystalloids, and typical crystalloid-associated pigment clusters were absent (*n* = 50; Fig 3E), as was observed in NTH-KO ookinetes. NTHΔPP parasites formed normal numbers of oocysts in mosquitoes (Figs 3B and EV3A), but like NTH-KO parasites these failed to produce sporozoites (1,000 oocysts examined

across at least 10 midguts from two independent infections) and had increased size (Figs 3D and EV3B). These collective findings indicate that the truncated NTH protein expressed in NTHΔPP parasites is structurally compromised and dysfunctional, resulting in a loss-of-function phenotype.

To assess the localisation of the crystalloid protein LAP3 in parasite line NTHΔPP, the latter was genetically crossed with parasite line LAP3/mCherry (Fig 3F). The expression of the functional *nth* allele in heterokaryotic zygotes allows their normal developmental progression through sporogony, as shown previously for LAP null mutants [29,31]. Accordingly, LAP3/mCherry female gametes that are fertilised by NTHΔPP male gametes give rise to heterokaryotic polyploid oocysts that sporulate normally (Fig 3F). The resultant sporozoite populations were transmitted to naïve mice by mosquito bite and drug selected. Diagnostic PCR carried out on the ensuing blood-stage parasite infections showed the presence of the modified *nth* allele (Fig 3G). Resultant ookinete cultures contained a mixture of the two parental lines (NTHΔPP and LAP3/mCherry) as well as double mutants possessing modified alleles for both *nth* and *lap3* (Fig 3E). Subcellular distribution of LAP3::mcherry in double mutant ookinetes was dispersed rather than concentrated in spots (Fig 3E), demonstrating that the absence of structurally intact NTH prevents LAP3 from reaching the crystalloid, consistent with the observed block in crystalloid formation.

## NTH has an essential structural role in crystalloid biogenesis

Besides NTH null mutants, ookinetes devoid of LAP1, LAP3, or DHHC10 also lack crystalloids [11–13,21]. The effect of NTH knock-out on sporozoite formation could thus be caused by the failure to form crystalloids and not reflect a direct role of NTH in sporogony. To test this hypothesis, we created a parasite line expressing a structurally intact, but enzymatically inactive version of NTH. To do so, the highly conserved aspartic acid residue at position 500 was mutated to a lysine (Fig EV2). The equivalent point mutation in bacterial NTH abolishes both hydride transfer and proton-translocating activities [32]. Ookinetes of the resulting NTH functional knockout (named NTH/ND500LK; Fig EV2) displayed GFP fluorescence in discrete spots that co-localised with pigment clusters (*n* = 50; Fig 3H), indicative of normal crystalloid formation. This shows that NTH needs to be physically present and structurally intact, but not enzymatically active, to facilitate crystalloid biogenesis.

Crystalloid proteins are trafficked via the ER, but specific sorting signals for the organelle have not been identified [10]. The crystalloid is a short-lived organelle that forms in ookinetes by coordinated congregation of small ER-derived vesicles, a process that is dependent on the synthesis of some of its protein constituents [11–13]. This indicates that crystalloid proteins are delivered to the organelle concurrent with its formation. One plausible explanation for the structural role of NTH in crystalloid biogenesis is that it interacts in the ER with other proteins destined for the organelle (e.g. LAPs), thereby ensuring that they are trafficked together and eliminating the need for individual proteins to possess specific crystalloid targeting signals. Indeed, interactions between LAPs have been shown to occur before crystalloid formation [21,33]. The formation of such a "crystalloid protein complex" also offers an explanation how several structurally and functionally unrelated crystalloid proteins including

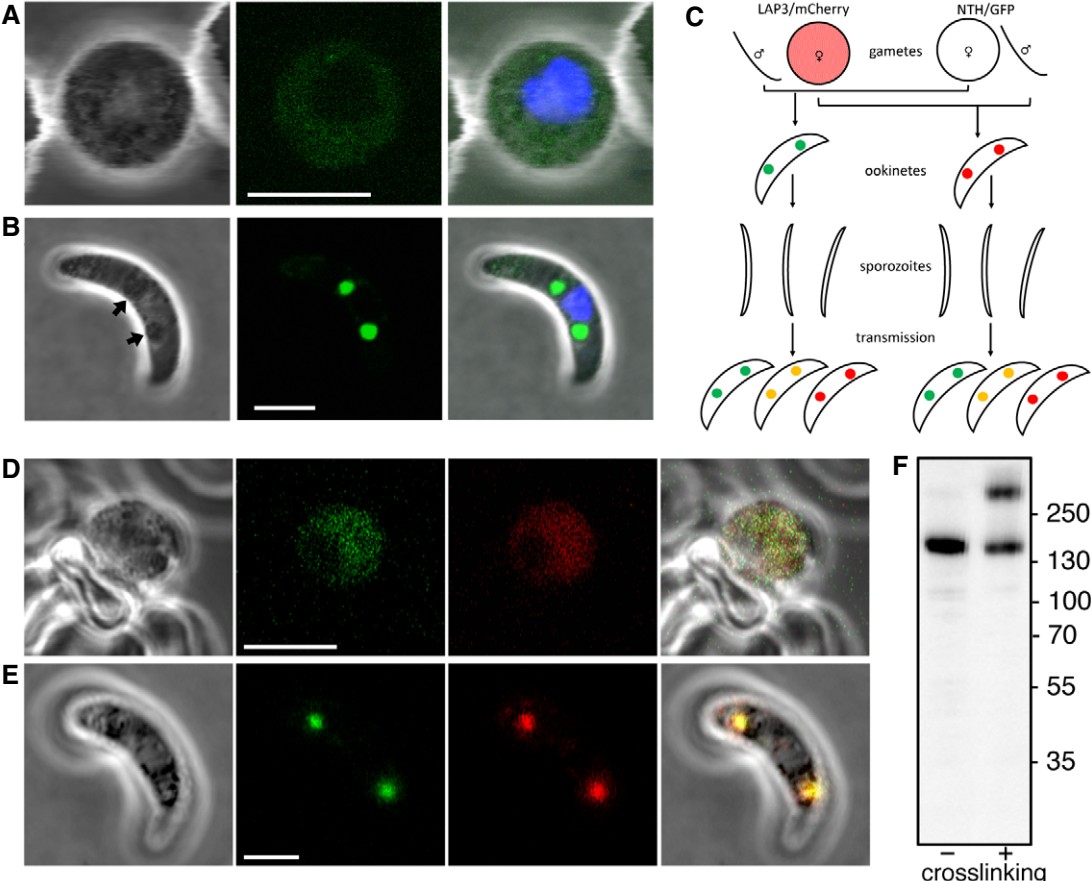

**Figure 2.  Expression and subcellular localisation of NTH in *Plasmodium berghei* ookinetes.**

A   Brightfield and fluorescence images of a NTH/GFP zygote at 4 h post-fertilisation, showing dispersed extranuclear fluorescence. Hoechst DNA staining (blue) marks nucleus. Scale bar = 5 µm.

B   Image of an NTH/GFP ookinete, showing fluorescent spots co-localising with pigment clusters (arrows). Hoechst DNA staining (blue) marks nucleus. Scale bar = 5 µm.

C   Schematic diagram of a genetic cross between parasite lines NTH/GFP and LAP3/mCherry and the resultant ookinete population after mosquito transmission and drug selection (oocysts, liver, and blood-stage parasites not shown). Colours represent NTH::GFP (green), LAP3::mCherry (red), and their co-localisation (orange). Spots in ookinetes depict crystalloids. Self-fertilisation events are omitted.

D   Brightfield and fluorescence images of a zygote at 4 h post-fertilisation simultaneously expressing NTH::GFP (green) and LAP3::mCherry (red). Scale bar = 5 µm.

E   An ookinete simultaneously expressing NTH::GFP (green) and LAP3::mCherry (red), showing co-localisation in the crystalloid. Scale bar = 5 µm.

F   Western blot of purified NTH/GFP ookinete samples with (+) and without (−) prior *in vivo* crosslinking. $M_r$ markers (kDa) are indicated on the right-hand side.

LAPs, DHHC10, and NTH produce very similar loss-of-function phenotypes [11–13]: Disruption of highly constrained interactions between crystalloid proteins, through their removal or structural alteration, could compromise formation of the crystalloids and consequently the downstream process of sporogony. Indeed, structural modifications of LAP family members have been shown to affect their ability to interact with each other [21] and impact on crystalloid biogenesis [11,12,21,28].

## NTH has an essential enzymatic role in sporogony

The normal crystalloid formation in parasite line NTH/ND500LK allowed us to assess the impact of crystalloids with enzymatically inactive NTH on sporogony. Like NTH structural knockout parasites, NTH/ND500LK parasites developed oocysts (Fig EV3A), confirming that NTH activity is not required for oocyst development *per*

*se*. Like the previous NTH mutants, NTH/ND500LK oocysts reached a larger size (Fig EV3B) and failed to sporulate (1,000 oocysts examined across at least 10 midguts from two independent infections), showing that NTH activity is essential for sporogony. The NTH functional knockout parasite is likely to have a normal protein repertoire in the organelle. The phenotype of this mutant parasite line can thus for the first time be pinpointed to the activity of a single crystalloid protein, identifying NTH as an essential molecule for crystalloid function, sporogony, and sporozoite transmission.

The predicted membrane topology of NTH projects its catalytic domains I and III inside the lumen of the crystalloid subunit vesicles, providing a source of luminal NADPH for the organelle. Indeed, the opposite topology would make little sense, because cytosolic NADPH/NADP$^+$ ratios are already high to maintain the reducing redox state of this environment. Our observation that NTH-deficient parasites are not impaired in oocyst formation

implies that the ookinetes and young oocysts have normal fitness and do not suffer extra from oxidative stress encountered in the insect. It is therefore plausible that crystalloid-resident NTH is not involved in maintaining the redox state, but instead facilitates specific NADPH-dependent anabolic reactions. In this context, the unusual structure of the crystalloid being a large cluster of small

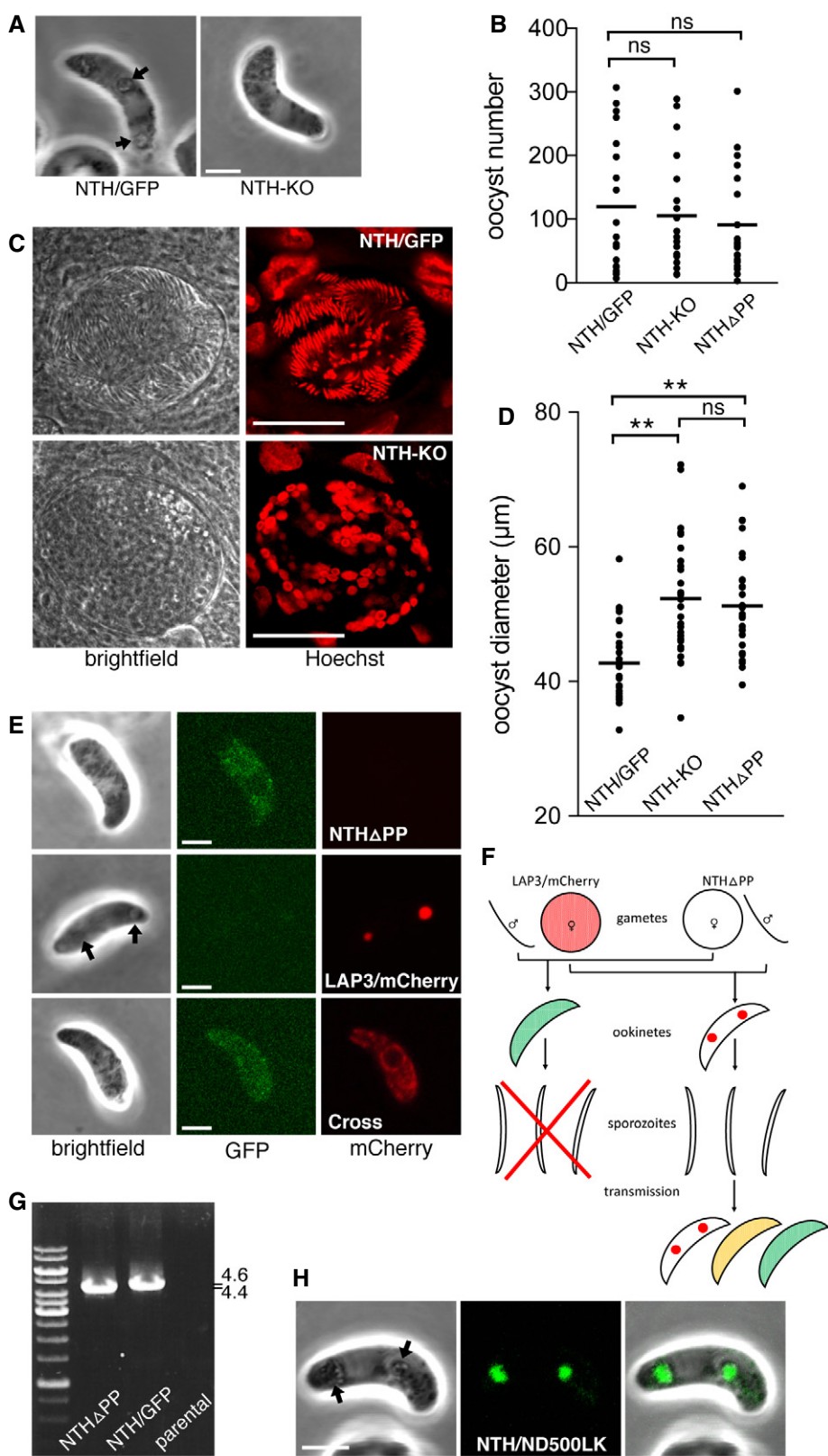

Figure 3.

**Figure 3.   NTH loss of function and contribution to *Plasmodium berghei* parasite development.**

A  Brightfield images of a NTH-KO and NTH/GFP ookinete, showing normal ookinete morphology, but lack of pigment clusters associated with crystalloids (arrows) in the NTH null mutant. Scale bar = 5 μm.

B  Aligned dot plot of oocyst numbers in *Anopheles stephensi* infected with parasite lines NTH/GFP, NTH-KO, and NTHΔPP. Horizontal lines mark mean values (*n* = 20); ns, not significantly different (Mann–Whitney).

C  Brightfield and fluorescence images of oocysts of parasite lines NTH/GFP and NTH-KO at 15 days post-infection, showing lack of sporulation in the NTH null mutant. Hoechst DNA staining (artificially depicted red) labels parasite nuclei. The large nuclei of the epithelial midgut cells also show. Scale bar = 20 μm.

D  Aligned dot plot of oocyst diameters in *A. stephensi* at 15 days post-infection with parasite lines NTH/GFP, NTH-KO, and NTHΔPP. Horizontal lines mark mean values (*n* = 20). ** mark statistically significant differences (*P* < 0.0001, Mann–Whitney); ns, not significant.

E  Images of ookinetes of parasite lines NTHΔPP showing dispersed NTHΔPP::GFP fluorescence; LAP3/mCherry showing focal LAP3::mCherry fluorescence in crystalloids; and cross (carrying modified alleles encoding NTHΔPP::GFP and LAP3::mCherry), showing dispersed LAP3::mCherry fluorescence. Arrow marks pigment cluster associated with crystalloids. Scale bar = 5 μm.

F  Schematic diagram of a genetic cross between parasite lines NTHΔPP and LAP3/mCherry and the resultant ookinete population after mosquito transmission and drug selection (oocysts, liver, and blood-stage parasites not shown). Colours represent NTH::GFP (green), LAP3::mCherry (red), and their co-localisation (orange). Spots in ookinetes depict crystalloids. The red cross indicates that sporozoite formation is blocked. Self-fertilisation events are omitted.

G  PCR diagnostic for the presence of modified *nth* alleles in blood-stage infections obtained after sporozoite transmission of a NTHΔPP x LAP3/mCherry genetic cross (lane NTHΔPP), amplifying an ~ 4.4 kb fragment. NTH/GFP parasites are used as a PCR control, amplifying ~ 4.6 kb. Parental parasites provide a negative control.

H  Image of a NTH/ND500LK ookinete, showing fluorescent spots co-localising with pigment clusters (arrows) indicative of normal crystalloid formation. Scale bar = 5 μm.

vesicles rather than a single large membrane-limited compartment could have biological relevance as it greatly increases the membranous area of the organelle and thus the amount of NTH activity, and that of other membrane-bound enzymes such as DHHC10, that can be accommodated.

## NTH is targeted to the sporozoite apicoplast

Apart from very young oocysts that still possess crystalloids, no discernible GFP fluorescence was observed in oocysts before sporulation (Fig 4A; *n* = 1,000). However, GFP fluorescence was observed in sporulated oocysts (Fig 4B) and in individual sporozoites (Fig 4C), indicating that NTH is once again expressed in sporozoites. Sporozoite NTH was concentrated in a tubular structure reminiscent of the apicoplast (Fig 4C), consistent with NTH possessing a predicted apicoplast transit peptide (Fig 1). To confirm apicoplast targeting of NTH, we co-localised it with the apicoplast-resident acyl carrier protein (ACP). To achieve this, ACP was fused to the red fluorescent protein mCherry and the resulting parasite line (named ACP/mCherry; Fig EV2) was genetically crossed with parasite line NTH/GFP. As ACP::mCherry was driven from a sporozoite-specific promoter, resultant heterokaryotic polyploid oocysts produced sporozoites that simultaneously expressed NTH::GFP and ACP::mCherry that localised to the same tubular structure (Fig 4D and E). Co-localisation of green and red fluorescence in the organelle was partial (Pearson's correlation coefficient: 0.66 ± 0.013, *n* = 21), pointing to their presence in distinct apicoplast compartments and possibly reflecting the predicted localisation of NTH and ACP in membrane and stroma, respectively. Finally, co-staining of NTH/GFP sporozoites with MitoTracker Red confirmed that NTH was absent from the mitochondrion (Fig 4F). The clear spatial separation of apicoplast and mitochondrion (Fig 4F) confirms previous observations that these two organelles are not physically connected in sporozoites [34].

In contrast to crystalloids, the apicoplast is present in the parasite throughout the life cycle, and nuclear-encoded apicoplast proteins are delivered in vesicles to a ready-formed organelle [35,36]. Protein trafficking to the apicoplast also occurs via the ER [37] and involves a bipartite amino-terminal signal composed of an amino-terminal ER signal peptide followed by an apicoplast

transit peptide [15,16], both of which are present in NTH (Fig 1). Although the sporozoite-specific expression of ACP::mCherry in our ACP/mCherry line (Fig EV2) precluded assessment of co-localisation with NTH::GFP in zygotes/ookinetes by genetic crossing, GFP fluorescence alone did not show evidence for a localisation of NTH in the apicoplast of these life stages (Fig 2). This indicates that in these life stages crystalloid targeting overrides apicoplast targeting. One explanation is that NTH is delivered to the crystalloid from the ER already in complex with the LAPs and/or other crystalloid proteins, preventing its trafficking to the apicoplast in the same cell. Because crystalloids are not present in sporozoites, the apicoplast becomes the default target organelle for NTH in this life cycle stage.

## NTH contributes to the transition from sporozoite to intraerythrocytic parasite

To assess the contribution of apicoplast-resident NTH to sporozoite infectivity in the vertebrate host, parasite line NTH/ND500LK was crossed with wild-type parasites to allow the formation of NTH/ND500LK sporozoites in heterokaryotic oocysts. Because the apicoplast targeting signal is unaltered in the NTH/ND500LK mutant (Fig EV2), the enzymatically inactive NTH::GFP protein in this parasite line is normally targeted to the apicoplast (Fig EV4A), making the sporozoites distinguishable from wild-type sporozoites present in the same mosquito. Mutant sporozoites were harvested from salivary glands, counted, and injected intravenously into groups of mice in parallel with mice injected with similarly obtained NTH/GFP sporozoites. Following inoculation, mice were pyrimethamine treated to prevent wild-type parasite development. Sporozoites of the enzymatically inactive NTH mutant could establish intraerythrocytic infection in the mouse (Fig EV4B, Table 1), indicating that NTH is not essential for the transition from sporozoite to blood-stage parasite. This is consistent with the successful transmission of NTHΔPP sporozoites upon crossing with LAP3/mCherry (Fig 3E and G). Nonetheless, NTH/ND500LK sporozoites were significantly less effective (*P* = 0.0029, Fisher) than their NTH/GFP counterparts in establishing blood-stage infections (Table 1), indicating that NTH activity makes an important contribution in the vertebrate host to sporozoite transmission.

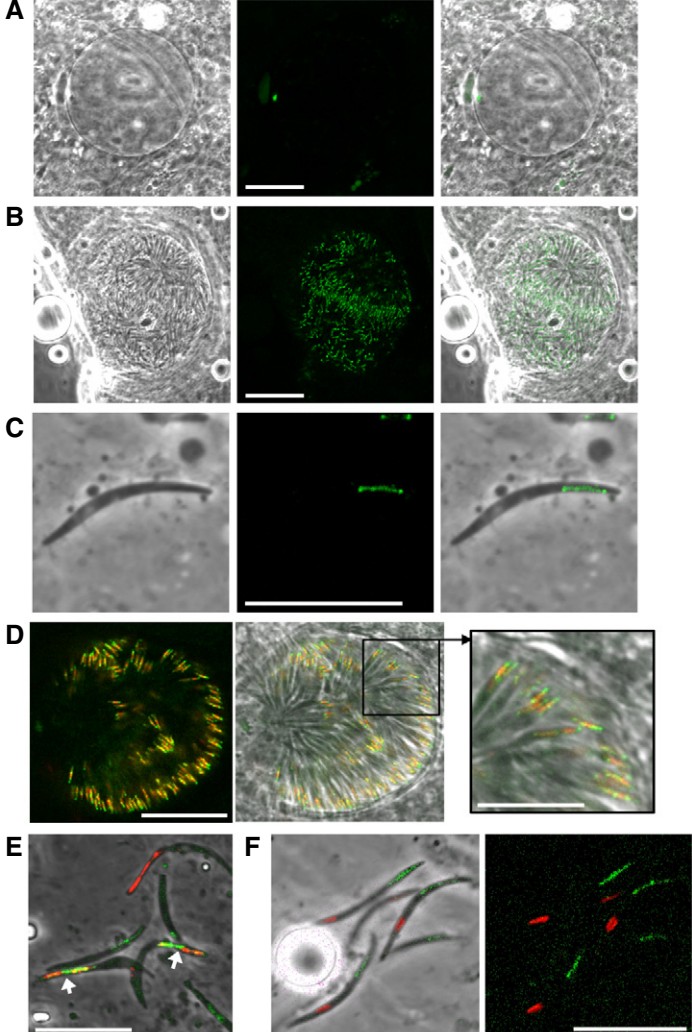

**Figure 4. Subcellular localisation of NTH in *Plasmodium berghei* oocysts and sporozoites.**

A   Brightfield and fluorescence image of an unsporulated NTH/GFP oocyst at 15 days post-infection. Scale bar = 20 μm.

B   Image of a sporulated NTH/GFP oocyst at 15 days post-infection. Scale bar = 20 μm.

C   Image of a NTH/GFP sporozoite, showing NTH::GFP localisation in a tubular structure. Scale bar = 10 μm.

D   Image of a sporulated oocyst simultaneously expressing NTH::GFP (green) and ACP::mCherry (red), showing co-localisation in the apicoplast. Scale bar = 20 and 10 μm (inset).

E   Image of sporozoites expressing NTH::GFP (green) and ACP::mCherry (red), showing co-localisation in the apicoplast (arrows). Sporozoites expressing only ACP::mCherry or NTH::GFP are also present. Scale bar = 10 μm.

F   Image of group of NTH/GFP sporozoites stained with MitoTracker Red CMXRos, showing distinct subcellular positions of apicoplasts (green) and mitochondria (red). Scale bar = 10 μm.

The *Plasmodium* apicoplast is a known site of several anabolic processes including biosynthesis of type II fatty acids (FASII) [38,39]. Several enzymes involved in its biosynthetic processes are NADPH-dependent, rendering the NADPH concentration potentially rate limiting. NADPH is also required for the generation of reduced ferredoxin by the apicoplast-resident ferredoxin-NADP$^+$ reductase [40,41]. However, the question what provides the source of NADPH in the apicoplast has remained unresolved. One reported hypothesis is that NADH, which can be formed via the decarboxylation of pyruvate to acetyl-coenzyme A by the apicoplast-resident pyruvate dehydrogenase complex, is able to replace NADPH as a reductant [38].

Based on the results from this study, we propose that the sporozoite-specific expression of NTH and its recruitment to the apicoplast serves to generate NADPH from this NADH source, increasing the biosynthetic activity of the organelle to meet the specific requirements of the sporozoite and possibly also its downstream hepatocytic life stages. In this context, it is noteworthy that FASII is redundant in blood and mosquito stage parasites, but is essential for liver stage development in rodent malaria species [42–45]. Moreover, and possibly for the same reason, several enzymes of FASII are uniquely found in sporozoite and liver stage apicoplasts [43]. However, in the human malaria parasite *Plasmodium falciparum*,

**Table 1. Ability of NTH/ND500LK and NTH/GFP sporozoites to cause blood-stage parasite infection in mice upon intravenous injection.**

| Experiment | Parasite line | No. of sporozoites injected | No. of mice infected/injected | Prepatent period (days)[a] |
|---|---|---|---|---|
| 1 | NTH/ND500LK | 100 | 0/5 | – |
| | NTH/GFP | 100 | 2/3 | 6.5 |
| 2 | NTH/ND500LK | 1,000 | 1/5 | 6 |
| | NTH/GFP | 1,000 | 4/4 | 5.5 |

[a]Average number of days after sporozoite injection until blood-stage parasites are first detected by microscopic examination of Giemsa-stained blood films.

FASII is required for sporozoite development [46] and it is possible that NTH, too, could function differently in this parasite species.

The presence of NTH in the *Plasmodium* crystalloid and apicoplast was surprising given its mitochondrial localisation in multicellular eukaryotes. *Plasmodium* spp. possess a mitochondrion-resident isocitrate dehydrogenase enzyme that is NADP-dependent, which could provide an alternative source of NADPH in this organelle [47,48] allowing the repurposing of NTH to other cell compartments. An apicoplast-specific *nth* gene in *Plasmodium* could have arisen through gene duplication and neo-specialisation followed by loss of the mitochondrial *nth* copy. This theory is supported by the presence of multiple *nth* genes in genomes of other apicomplexan parasites. Subsequent mutations, for example, promoting physical interactions with crystalloid proteins, may then have led to its acquisition by the crystalloid organelle. Stage-specific targeting of the same gene product to distinct organelles precludes fundamental changes of the protein's structure, suggesting that NTH fulfils functionally identical roles in each of the organelles albeit in a different subcellular environment and molecular context.

Like *Plasmodium* NTH, proton-translocating NTH proteins described in the protozoan parasites *Eimeria tenella* and *Entamoeba histolytica* have βα architectures [2]. In these organisms, too, the enzyme appears to be absent from mitochondria or functionally related organelles. Instead, NTH has been shown to localise to membranes of phagosomes and lysosomes in *Entamoeba* [49], whilst it has been localised in the refractile body in *Eimeria* [50]. The refractile body is a structure of unknown function that has been postulated to be functionally similar to the *Plasmodium* crystalloid [51]. Thus, a picture is emerging that unicellular eukaryotes utilise NTH in a range of organelles. This reveals a much wider role of membrane-bound NTH in eukaryotic cell biology than previously assumed, beyond a function in the mitochondrion. The functional dependence of the crystalloids on NTH as shown here warrants further studies aimed at identifying the underlying NADPH-dependent processes in this organelle and their intriguing role in sporogony.

## Materials and Methods

### Animal use

All laboratory animal work was carried out in accordance with the United Kingdom Animals (Scientific Procedures) Act 1986 implementing European Directive 2010/63 for the protection of animals used for experimental purposes and was approved by the London School of Hygiene & Tropical Medicine ethical review committee and United Kingdom Home Office. Experiments were generally conducted in 6- to 8-week-old female CD1 or C57Bl/6 mice obtained from established breeders, specific-pathogen-free, and maintained in filter cages. Animal welfare was assessed daily, and animals were humanely killed upon reaching experimental or clinical endpoints. Mice were infected with parasites suspended in phosphate-buffered saline (PBS) by intraperitoneal injection or by infected mosquito bite on anaesthetised animals. Intraerythrocytic parasitemia was monitored regularly by collecting of a small volume of blood from a superficial tail vein. Drugs were administered by intraperitoneal injection or where possible were supplied in drinking water. Parasitised blood was harvested by cardiac bleed under general anaesthesia without recovery.

### Parasite maintenance, culture, purification, and transmission

*Plasmodium berghei* ANKA clone 2.34 parasites were maintained as cryopreserved stabilates or by mechanical blood passage and regular mosquito transmission. Mosquito infection and transmission assays were as previously described using *A. stephensi* [52,53], and infected insects were maintained at 20°C at ~ 70% relative humidity under a 12-h/12-h light/dark cycle. Ookinete cultures were set up overnight from gametocytemic blood as previously described [54].

### Generation of NTH constructs

The elongation factor 1α promoter was PCR amplified from *P. berghei* genomic DNA with primers EFpromoter-F2 (GCATGCCT GCAGGTCAAATGGCAGTTATTAAAAAATAAGG) and EFpromoter-R2 (AACCAACCATGAATTCTTTTTATATATTTATACACAAAGTATA TATATTTTAGG) and introduced by in-fusion into HincII/EcoRI-digested pLP-hDHFR/EcoRI [55] to give pLP-hDHFR/EF1α. Following this, the deltaIMC1b/EGFP sequence was PCR amplified from plasmid pDNR-deltaIMC1b/EGFP [56] with primers pBS-hDHFR-F (TAGGGCGAATTGGGCTGCAGTCGACGGTACCATTGAGACG) and pBS-hDHFR-R (GAAAAGAATTAAGCTGGCGCCGAAATTGAAGGAA AAAACATC) and introduced by in-fusion into PstI/HindIII-digested pLP-hDHFR/EF1α to create plasmid pBS-EGFP-hDHFR. A ~ 4.5 kb fragment corresponding to the coding sequence and 5′UTR of PBANKA_1317200 (*nth*) was PCR amplified form *P. berghei* genomic DNA with primers P25-F (TTGGGCTGCAGTCGAGG TACCGTTTAGTAGACATATAAAAATGACGTGTACATAC) and P45 10-R (ATGAGGGCCCCTAAGCTCTTTACAAACATATCAAGCATTCT TTTTG) and introduced into SalI/HindIII-digested pBS-EGFP-hDHFR by in-fusion to give plasmid pBS-NTH2/GFP. A ~ 0.6 kb fragment corresponding to the 3′ UTR of PBANKA_1317200 was PCR amplified with primers P4560-F (ATATGCTAGAGCGGCCTGTACTT GATTTCACTATTTATTATATAGAACC) and P5170-R (CACCGCGG

TGGCGGCCGATCATTTTCAAGTGTCCCTAC) and introduced into *NotI*-digested pBS-NTH2/GFP by in-fusion to give pBS-NTH3/GFP. This construct was used to GFP-tag NTH at the carboxy-terminus to generate parasite line NTH/GFP.

To generate a construct for *nth* gene knockout, pBS-NTH3/GFP was PCR amplified with primers NTH-KO-R (CTAAGCTTCCTA CAATGGTTATGTCCAATCAGATG) and NTH-KO-F (ATTGTAG GAGCTTAGGGGCCCTCAT), and the PCR product was circularised by in-fusion to give pBS-NTH-KO. This removes the entire *nth* sequence except for the first 21 codons, in frame with the *gfp* module, to generate parasite line NTH-KO.

To generate a construct removing residues 25–83 from the *nth* coding region, pBS-NTH3/GFP was PCR amplified with primers NTHdeltaPP-F (TTAAATCTCGAGGCTCCGTCTTATTCGTTCATACC) and NTHdeltaPP-R (CTCGAGATTTAATCTACAATGGTTATGTC CAATC), and the product was circularised by in-fusion to give pBS-NTH4/GFP. This introduces a diagnostic XhoI restriction site at the site of the deletion. This plasmid was used to generate parasite line NTHΔPP.

To generate a construct containing an enzymatically inactive *nth* coding region, pBS-NTH3/GFP was PCR amplified with primers NTH-ND500LK-F (AGCGCTTAAAATTATCAATCCATCTTCCTTAGA TCC) and NTH-ND500LK-R (ATAATTTTAAGCGCTCCAACAAC TAAAACTAAATCAAC), and the product was circularised by in-fusion to give pBS-NTH5/GFP. This introduces a diagnostic AfeI restriction site at the site of the point mutation. This plasmid was used to generate parasite line NTH/ND500LK.

### Generation of ACP::mCherry construct

To tag ACP with mCherry, the entire *acp* coding sequence (plus introns) was PCR amplified with primers ACP-mCherry-F (ATTTTTTTCCCATTTAAAGCTTATGCTGTTTCGCAAAATGAA) and ACP-mCherry-R (GAGGGCCCCTAAGCTTGCATCAGGCTTTTTATTT TTTTC) and cloned into PacI/HindIII-digested pLP-IMC1a/mCherry [55] to give pLP-ACP/mCherry. In the resultant parasite line ACP::mCherry, expression is driven from the sporozoite-specific *imc1a* promoter.

### Generation and genotyping of parasite lines

Plasmids were linearised with KpnI and SacII prior to gene targeting by double crossover homologous recombination. Parasite transfection, pyrimethamine selection, and limiting dilution cloning were performed as previously described [57,58]. Genomic DNA extraction for diagnostic PCR was performed as previously described [53].

### Microscopy

Live parasite samples were assessed, and images captured, on a Zeiss LSM510 or LSM880 laser scanning confocal microscope using 100× oil objective and ZEN 3.0 software.

**Expanded View** for this article is available online.

### Acknowledgements

We thank Elizabeth McCarthy for assistance with the microscopy and Graham Clark for critical reading of the manuscript. This research was jointly funded by the UK Medical Research Council (MRC) and the UK Department for International Development (DFID) under the MRC/DFID Concordat agreement (reference MR/P021611) and by grants from the Wellcome Trust (reference 088449) and the UK Biotechnology and Biological Sciences Research Council (reference BB/M001598).

### Author contributions

The project was conceived by JTD; experiments were planned by SS, AZT, and JTD; experiments were performed by SS, AZT, VS, EL, and JTD; data were analysed and interpreted by SS, AZT, EL, and JTD; and the manuscript was written by JTD with comments from all authors.

### Conflict of interest

The authors declare that they have no conflict of interest.

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
