## [Review Process File · EMBO Reports]

NAD(P) transhydrogenase has vital non-mitochondrial functions in malaria parasite transmission

Sadia Saeed, Annie Z. Tremp, Vikram Sharma, Edwin Lasonder and Johannes T. Dessens

Review timeline:

Submission date:	30 January 2019
Editorial Decision:	5 March 2019
Revision received:	15 November 2019
Editorial Decision:	9 December 2019
Revision received:	11 December 2019
Accepted:	17 December 2019

Editor: Achim Breiling

Transaction Report:

1st Editorial Decision

5 March 2019

Thank you for the submission of your research manuscript to EMBO reports. We have now received the reports from the three referees that were asked to evaluate your study, which can be found at the end of this email.

As you will see, all referees think the manuscript is of interest, but requires major revisions to allow publication. As the reports are below, I will not detail them here, but I think it will be essential to show that Plasmodium NTH is important for sporozoite formation due to its role in the crystalloid body or in the apicoplast, as indicated by referee #3, who also suggests experiments to address this. Also points made by referees #1 and #2 regarding reproducibility, quantifications and statistics need particular attention.

Given the constructive referee comments, we would like to invite you to revise your manuscript with the understanding that all referee concerns must be addressed in the revised manuscript and in a detailed point-by-point response. Acceptance of your manuscript will depend on a positive outcome of a second round of review. It is EMBO reports policy to allow a single round of revision only and acceptance or rejection of the manuscript will therefore depend on the completeness of your responses included in the next, final version of the manuscript.

Revised manuscripts should be submitted within three months of a request for revision; they will otherwise be treated as new submissions. Please contact me if a 3-months time frame is not sufficient so that we can discuss the revisions further.

Supplementary/additional data: The Expanded View format, which will be displayed in the main HTML of the paper in a collapsible format, has replaced the Supplementary information. You can submit up to 5 images as Expanded View. Please follow the nomenclature Figure EV1, Figure EV2 etc. The figure legend for these should be included in the main manuscript document file in a section called Expanded View Figure Legends after the main Figure Legends section. Additional Supplementary material should be supplied as a single pdf labeled Appendix. The Appendix includes a table of content on the first page, all figures and their legends. Please follow the nomenclature Appendix Figure Sx throughout the text and also label the figures according to this

nomenclature.

For more details please refer to our guide to authors:

<http://embor.embopress.org/authorguide#manuscriptpreparation>

Important: All materials and methods should be included in the main manuscript file.

See also our guide for figure preparation:

http://www.embopress.org/sites/default/files/EMBOPress_Figure_Guidelines_061115.pdf

Please also follow our guidelines for the use of living organisms, and the respective reporting guidelines: <http://embor.embopress.org/authorguide#livingorganisms>

Further, please format the references according to EMBO reports style. See:

<http://embor.embopress.org/authorguide#referencesformat>

Please also add a conflict of interest statement, and a paragraph detailing the author contributions to your manuscript (above the acknowledgements).

Regarding data quantification and statistics, can you please specify, where applicable, the number "n" for how many independent experiments (biological replicates) were performed, the bars and error bars (e.g. SEM, SD) and the test used to calculate p-values in the respective figure legends. Please provide statistical testing where applicable. See: <http://embor.embopress.org/authorguide#statisticalanalysis>

We now strongly encourage the publication of original source data with the aim of making primary data more accessible and transparent to the reader. The source data will be published in a separate source data file online along with the accepted manuscript and will be linked to the relevant figure. To use this opportunity, please submit the source data (for example scans of entire gels or blots, data points of graphs in an excel sheet, additional images, etc.) of your key experiments together with the revised manuscript. Please include size markers for scans of entire gels, label the scans with figure and panel number, and send one PDF file per figure.

- a complete author checklist, which you can download from our author guidelines (<http://embor.embopress.org/authorguide#revision>). Please insert page numbers in the checklist to indicate where the requested information can be found.
- a letter detailing your responses to the referee comments in Word format (.doc)
- a Microsoft Word file (.doc) of the revised manuscript text
- editable TIFF or EPS-formatted single figure files in high resolution (for main figures and EV figures)

Please also note that we now mandate that the corresponding author lists an ORCID digital identifier that is linked to his/her EMBO reports account!

I look forward to seeing a revised version of your manuscript when it is ready. Please let me know if you have questions or comments regarding the revision.

REFeree REPORTS

Referee #1:

In their manuscript Saeed et al. present a genetic characterisation of the single NAD(P) transhydrogenase of *Plasmodium berghei*. Their major finding is that NTH localises to the crystalloid and in sporozoites to the apicoplast and that the protein functions both at a structural level helping to form the crystalloid organelle and enzymatically during sporulation. In particular the latter provides a nice example of the use of different genetics approaches to elucidate different functions of a single protein, thus going a step further compared with earlier findings about the functioning of other crystalloid proteins from the same lab. In general, the study is well performed

and makes a rather nice use of cross-fertilisation to generate double labelled offspring. However, in its current form I think that the manuscript lacks in species transcending or more systematic insights in its functioning in Plasmodium to warrant publication in EMBO Reports. The lines and stages used for imaging appear somewhat random and I fail to see the logic in their choices. A more systematic approach as suggested below would significantly strengthen the paper.

Major concerns:

- Page 4 & Fig.2: in the text the authors refer to a typical staining in two foci as shown in panel 2A, but next in the co-localisation with LAP3 in panel 2B they show only a single dot. If these also typically demonstrated dual targeting a representative image should be provided. However, the choice for this image raises my concern. Why did the authors choose this as a 'representative image'? Do the double-labelled parasites typically demonstrate a single locus for both proteins? Are the two proteins always co-localising? I can imagine that this is not the case if NTH localises to the crystalloid as well as the apicoplast while LAP3 does not. Also, if there are significant differences between the staining patterns of LAP3 and NTH this should be quantified, ideally on a clonal/isogenic line. The authors need to explain the observations in much more detail if there are discrepancies, or choose more representative images. Also they should use ACP to demonstrate co-localisation or absence thereof in the ookinete and oocyst.

- Page 5 & Fig.3: The images in panel 3A are difficult to interpret and to my mind do not univocally support the notion that crystalloids are not being formed. This is then later confirmed in the NTHdeltaPP line after crossing with the LAP3 marker, but I fail to see the reason why the authors did not perform this experiment with the complete knockout line instead. Quantification of the effect unless this is absolute would be a welcome addition. It is unclear whether the counted oocyst numbers are from a single experiment or not. This should be clearly indicated. Though commonly applied in the field, I'm not a big fan of combining individual counts or measurements per mosquito, but would rather see experimental averages of at least three completely independent experiments plotted and see the statistics performed on these data, rather than treating each mosquito as an individual experiment. This also applies to the data in panel 3D. In Panel 3E the authors again chose to show an ookinete with a single dot (see my previous comment). Given that the authors performed a cross-fertilisation experiment allowing them to bypass the detrimental effect during oocyst development, it would be a very interesting addition if they could explore whether the deletion of NTH has an effect on liver stage development (even though perhaps not expected) and quantify the effect if any, as done recently by Rathnapala et al 2017 and Matz et al 2018. This is even more relevant with regard to the observed apicoplast localisation of NTH in sporozoites and a potential role in this organelle during liver stage development as discussed by the authors.

- Page 6 & Fig.3: While the image in panel 3G looks convincing, the claim that the crystalloids are intact in the enzymatically inactive line would be further supported using the LAP3 marker once more. The oocyst phenotype of the enzymatic mutant should be quantified as for the structural mutants taking my comments on that quantification in consideration.

- Given the phenotypes of the various functional mutants, I'm surprised that the authors have not provided more data on the localisation of NTH, co-localisation with LAP3 and ACP, and localisation of LAP3 in the functional mutants at in eg zygotes and during different time points of oocyst development. I believe this would be a rather important addition to complete the story. The single non-sporulated oocyst shown is taken at day 15 and could also represent an unhealthy/dying oocyst, it would help to show earlier time points.

Additional minor points:

- Please add line numbers for ease of reviewing and commenting
- Abstract: only shown for Plasmodium, shouldn't extrapolate to all protozoans
- Page 3: the apicoplast is not a relic chloroplast but a separately derived plastid of likely red algal origin (Janouskovec et al 2010)
- Page 3: The claim of the final sentence is far too broad and not supported by the data, no data are presented for any other than Plasmodium spp and no other organelle than the crystalloid, which to the best of my knowledge has not been identified in any other spp outside the Haemosporina.
- Page 4: The dispersed NTH-GFP-fluorescence in zygotes (which should be shown, perhaps in the supplement) might be preassembly of crystalloid protein complexes. It would be interesting to confirm this theory of preassembly by investigating the zygote stages / early oocysts of the LAP3-

mCherry/NTH-GFP parasite in fluorescence microscopy and check whether they colocalize before assembly in the crystalloids.

- Page 5 & Fig.S1: It would be helpful to indicate in the schematic that the ATS has been removed in NTHdeltaPP. In general, a little more detail on the genetics would help to clarify how the genome has been manipulated exactly without having to refer to the methods and the primer sequences.

- Page 7 & Fig.4: The partial co-localisation of NTH and ACP could also be linked to their localisation in membrane vs. matrix. I'm surprised that the authors chose to generate a gene deletion mutant to integrate the ACP marker rather than to clone the desired promoter (or better still a more constitutive promoter) into the vector and integrate the vector elsewhere. Better options are available for the stable integration of transgenes. Furthermore, apicoplast and mitochondrion in Plasmodium parasites are closely associated both in intertwined metabolism and physical proximity, with sporozoites forming the exception to the dogma. As I do not think this is common knowledge it would help to refer to Kudryashev et al 2010 to explain this atypical organelle positioning and support the case for NTH targeting to the apicoplast.

- Page 7: Dual targeting, while not often, is observed for other apicoplast and mitochondrial proteins in Plasmodium. However, given the presented data I find the term dual targeting to be slightly misleading. In no life cycle stage NTH is targeted to two sites (at least based on the data currently presented, see my earlier comment). Stage-specific targeting would be a more suitable term in my opinion. A little speculation on the mechanism of stage-specific targeting would also be welcome. The apicoplast and the ATS are always present, so how could this be overruled to favour the crystalloid assembly/trafficking?

- Page 7: In the discussion, the authors completely fail to discuss the discrepancies in apicoplast function (in particular FasII) between *P. berghei* and *P. falciparum* (Van Schaijk et al 2014). Analogous to these differences, NTH could function rather differently in the different Plasmodium spp.

- A phylogenetic analysis of the various apicomplexan and other NTHs could provide support for the evolutionary history of the single Plasmodium copy.

- Page 8: There is no support whatsoever for the final statement of the manuscript, nor do the authors suggest in any way what such an intervention strategy should look like. This sentence should be omitted.

- In the methods there is no section describing how microscopy was performed.

Referee #2:

This paper describes a membrane-bound NAD(P) transhydrogenase (NTH) in the malaria parasite. This class of enzymes catalyzes hydride ion transfer between NAD(H) and NADP(H). In metazoans the enzyme is localized in the inner membrane of the mitochondrion. In this study the only malaria parasite NTH is described and found to be localized to the so called crystalloid, a Plasmodium specific organelle, which plays an important role in the development of the parasite in the mosquito. Specifically the crystalloid is needed in the sporogonic oocyst for development of the infectious sporozoites. Previously in a number of studies it has been shown that in the absence of the crystalloid the sporozoites do not develop and transmission is blocked. The authors generated a null mutant of the gene encoding NTH and show that the phenotype of this mutant also affects the development of the sporozoites. They also generated two other mutants in which the phenotype was similar to the null mutant. In the one mutant a truncated form of NTH, lacking an ER signal sequence, was expressed while in the second a point mutation was introduced, which abolished enzymatic activity. The formation of the crystalloid was also investigated in the three mutants and found to be aberrant in the first two mutants while normal in the third. Together these results suggest that NTH is necessary for formation of the crystalloid but its enzymatic activity is not essential for this process. In addition the authors localized the protein to the apicoplast organelle in the sporozoites. The conclusions are solid and supported by the experimental evidence.

This paper presents interesting and important finding as the NTHs in other eukaryotic organisms are mitochondrial enzymes while in the malaria parasite it has an essential role in the parasite-specific organelles crystalloid. It was also detected in the apicoplast, an organelle found in Apicomplexan parasites, though its function in this organelle was not investigated. The results will be interesting to researchers interested in mitochondrial enzymes and working with the mosquito stages of the malaria parasite, possibly also to those studying other Apicomplexan parasites.

I do have some reservations against publication in its present form with the major one being that the function in the sporozoite stage was not investigated. This is certainly not without its challenges and would require at least a few months to a year of work. However, the results would significantly enhance the impact of the paper.

The paper is clearly written and put in context with previous literature. However, some figures need improvement. For example a schematic figure of the generated mutants and the crossing strategy (Fig. 2B and 3F) would be very helpful. Also the oocyst pictures in Fig. 3C and 3H are not of good quality. Figure 3B is problematic. The number of counted midguts should be inserted. The authors should also repeat these experiments at least once.

Referee #3:

This is an interesting, although yet incomplete study, on the sole NTH enzyme (produces NADPH) in *Plasmodium*. If completed, this could also be of interest to scientists outside the malaria research community as it is reported that this enzyme localises to two distinct organelles. It is unfortunately not made clear whether the enzyme plays an essential role in each of these organelles but this can be easily addressed (see my comments below). These organelles are found in two different life cycle forms of the parasite, the ookinete that infects the mosquito and the sporozoite, which is transmitted by the mosquito. In the ookinete NTH localises to the crystalloid bodies (specific to ookinetes) and in sporozoite to the apicoplast (found in all parasite forms). Through reverse genetic approaches it is shown by gene deletion, domain deletion and point mutation that the protein is essential for crystalloid body and sporozoite formation.

What remains however unclear, is whether the enzyme is important for sporozoite formation due to its role in the crystalloid body or in the apicoplast. This should be experimentally addressed prior to resubmission.

One way such an experiment could be conducted is to cross the existing NTH-GFP line with (1) an NTHdPP-mCherry line and (2) an NTH/ND500LK-mCherry line. This should yield oocysts of mixed color that produce haploid sporozoites. These should be examined for the localization of the NTH and for their infectivity to mice. If the NTH/ND500LK mutation leads to the expected drop of NADPH in the apicoplast, these sporozoites should fail to infect mice and hence the resulting blood stage population should all (or at least a much larger proportion than the input sporozoite population) be NTH-GFP parasites. Similarly, if the NTHdPP fails to localize to the apicoplast, these sporozoites should also fail to infect mice and hence the resulting blood stage population should again all be NTH-GFP parasites. This is an essential experiment to also determine if the enzyme is a potential drug target for malaria prevention.

In addition the paper would benefit from a series of light and electron microscopy images that investigate the difference in sporogony between the NTH-KO and the wild type parasites as well as the localization of the fluorescence of the NTH/ND500LK mutant during sporogony.

Minor comments:

The paper would also benefit if it would include a life cycle to make it easier for a non-specialist to understand the different forms of the parasite.

Page 3: 13th line of introduction contains a 'they' that should be deleted

page 5: lines 5 and 6: please rephrase this statement, the higher molecular weight band could also arise due to an interaction with another protein, hence it is not 'most likely' a homodimer, but 'possibly' a homodimer. This 'could be' consistent with reports...

page 8: add a note that at least hints to the possibility that the refractile body in *Eimeria* is somewhat similar to the crystalloid body in *Plasmodium*.

Explain how the knowledge generated in this paper could lead to novel malaria transmission control

strategies.

Figure 2B, 3E: is there only one crystalloid body in the double fluorescent parasites?

Figure 1 and 2 could easily be fused to accommodate the necessary new data.

1st Revision - authors' response

15 November 2019

Referee #1:

In their manuscript Saeed et al. present a genetic characterisation of the single NAD(P) transhydrogenase of *Plasmodium berghei*. Their major finding is that NTH localises to the crystalloid and in sporozoites to the apicoplast and that the protein functions both at a structural level helping to form the crystalloid organelle and enzymatically during sporulation. In particular the latter provides a nice example of the use of different genetics approaches to elucidate different functions of a single protein, thus going a step further compared with earlier findings about the functioning of other crystalloid proteins from the same lab. In general, the study is well performed and makes a rather nice use of cross-fertilisation to generate double labelled offspring. However, in its current form I think that the manuscript lacks in species transcending or more systematic insights in its functioning in *Plasmodium* to warrant publication in EMBO Reports. The lines and stages used for imaging appear somewhat random and I fail to see the logic in their choices. A more systematic approach as suggested below would significantly strengthen the paper.

Major concerns:

- Page 4 & Fig.2: in the text the authors refer to a typical staining in two foci as shown in panel 2A, but next in the co-localisation with LAP3 in panel 2B they show only a single dot. If these also typically demonstrated dual targeting a representative image should be provided. However, the choice for this image raises my concern. Why did the authors choose this as a 'representative image'. Do the double-labelled parasites typically demonstrate a single locus for both proteins? Are the two proteins always co-localising? I can imagine that this is not the case if NTH localises to the crystalloid as well as the apicoplast while LAP3 does not. Also, if there are significant differences between the staining patterns of LAP3 and NTH this should be quantified, ideally on a clonal/isogenic line. The authors need to explain the observations in much more detail if there are discrepancies, or choose more representative images. Also they should use ACP to demonstrate co-localisation or absence thereof in the ookinete and oocyst.

The presence of a single fluorescent spot/crystalloid in the images in question was pure coincidence. We have shown previously that mature *P. berghei* ookinetes possess between 1 and 3 crystalloids [1]. In addition, the confocal microscopy is carried out on live ookinetes in suspension and it can be difficult to capture more than one crystalloid in the same focal plane. This is why the images in Fig 2B and 3E showed only a single spot/crystalloid; the point of the images was to show the (co)-localization of LAP3::mcherry, not how many crystalloids are present. However, we have now replaced these images with ones that show two spots/crystalloids. LAP3 and NTH are encoded by single copy genes located on distinct chromosomes, allowing double mutants to be obtained from a genetic cross. We did not observe differences in the localisation patterns of LAP3 and NTH; instead LAP3::mCherry and NTH::GFP always co-localise, both before crystalloid formation in zygotes and after crystalloid formation in ookinetes. This is shown in the revised Figure 2 (D and E).

We could not use our ACP/mCherry line to visualise the ookinete's apicoplast, as it uses a sporozoite-specific promoter to drive ACP::mCherry expression. We did nonetheless look carefully at whether NTH was present in the zygote/ookinete apicoplast which, unlike crystalloids, is visible as a single tubular structure [2]. First, in at least 50 ookinetes examined, we found no evidence that NTH::GFP was present in such a structure (e.g. Fig. 2BE). Second, in 4h zygotes the crystalloids are not yet formed [1], but an intact apicoplast is already present. Thus, targeting of NTH::GFP to the apicoplast would result in a localised signal in this life cycle stage. The fact that we observed NTH::GFP only as diffuse extranuclear fluorescence (Fig. 2AD) indicates that NTH is not targeted to the apicoplast. Furthermore, apicoplast localisation in zygotes cannot be easily masked by, or mistaken for, crystalloid localisation. Third, LAP3, which is not an apicoplast protein and does not possess an apicoplast transit peptide, perfectly colocalised with NTH in cells that express both these transgenes (Fig. 2DE). These collective data provide evidence that NTH is absent from the apicoplast in ookinetes and upstream zygotes. We have now added a sentence to the manuscript to specifically point this out, along with a possible explanation for why this is the case (lines 271-275).

- Page 5 & Fig.3: The images in panel 3A are difficult to interpret and to my mind do not univocally support the notion that crystalloids are not being formed. This is then later confirmed in the NTHΔPP line after crossing with the LAP3 marker, but I fail to see the reason why the authors did not perform this experiment with the complete knockout line instead. Quantification of the effect unless this is absolute would be a welcome addition.

We did not cross LAP3/mCherry with the complete knockout (NTH-KO) line, because green fluorescence would reflect the localisation of GFP rather than NTH. This is why we generated parasite line NTH Δ PP that expresses a truncated NTH protein. We did not observe crystalloids in any ookinetes (50 ookinetes examined), indicating that the effect of NTH disruption on crystalloid formation is absolute, as is indeed the case for LAP1, LAP3 and DHHC10 knockout parasites [1, 3, 4]. We now mention in the text how many cells were examined.

It is unclear whether the counted oocyst numbers are from a single experiment or not. This should be clearly indicated. Though commonly applied in the field, I'm not a big fan of combining individual counts or measurements per mosquito, but would rather see experimental averages of at least three completely independent experiments plotted and see the statistics performed on these data, rather than treating each mosquito as an individual experiment. This also applies to the data in panel 3D.

The oocyst numbers in Fig. 3B are from a single experiment in which the different parasite lines were directly compared. This figure was included to make the point that NTH is not involved in ookinete infectivity and oocyst formation (expressed as oocyst numbers), as has been shown for other crystalloid proteins (e.g. the LAPs and DHHC10, [1, 3-5]). In our view, this is best demonstrated by directly comparing the wildtype control (NTH/GFP) with the structural NTH knockout (NTH-KO). Given that complete structural knockout of NTH has no effect on oocyst numbers (Fig. 3B), the inclusion of the NTH Δ PP mutant parasite in Fig. 3B in effect already confirms the lack of NTH involvement in oocyst formation. Nonetheless, to show further reproducibility of the results we have repeated oocyst counts in an independent experiment comparing the different lines directly, and this has been added to the manuscript as expanded view Fig. EV3A. A similar thing was done for Fig. 3D regarding oocyst diameters (Fig. EV3B). In our experience, it is harder to compare oocyst numbers between experiments due to inter-experiment variability, which can cause overall infection levels to differ considerably from one mosquito experiment to the next.

In Panel 3E the authors again chose to show an ookinete with a single dot (see my previous comment).

This image has now been replaced with one showing two spots, see our earlier comment.

Given that the authors performed a cross-fertilisation experiment allowing them to bypass the detrimental effect during oocyst development, it would be a very interesting addition if they could explore whether the deletion of NTH has an effect on liver stage development (even though perhaps not expected) and quantify the effect if any, as done recently by Rathnapala et al 2017 and Matz et al 2018. This is even more relevant with regard to the observed apicoplast localisation of NTH in sporozoites and a potential role in this organelle during liver stage development as discussed by the authors.

We have indeed successfully used cross-fertilization of parasite line NTH Δ PP (which does not form crystalloids nor sporozoites) with a parasite that contains a functional *nth* allele (in this case LAP3/mCherry) to bypass the block in sporogenesis in the oocysts and produce NTH Δ PP sporozoites. The fact that these sporozoites are transmissible by mosquito bite, resulting in NTH Δ PP blood stage parasites as shown in Fig. 3E and G, already indicates that functional NTH is not essential for liver stage development of the parasite. To further explore the contribution of NTH to transition from sporozoite to blood-stage parasite, we have carried out and included new experiments investigating the efficacy of NTH/ND500LK sporozoites to cause blood stage infections relative to control (NTH/GFP) sporozoites (lines 278-293, Fig. EV4, Table 1). Briefly, sporozoites from each population were obtained by crossing the mutants with wildtype (parental) *P. bergeri* parasites. Sporozoites of each GM parasite (recognisable from having a green apicoplast) were then harvested from salivary glands, counted, and known numbers injected intravenously into groups of naive mice. The results obtained show that NTH/ND500LK sporozoites are significantly less effective than their NTH/GFP counterparts in causing intraerythrocytic parasite infection (Table 1), indicating that functional NTH contributes to parasite development in the liver.

- Page 6 & Fig.3: While the image in panel 3G looks convincing, the claim that the crystalloids are intact in the enzymatically inactive line would be further supported using the LAP3 marker once more.

Ookinetes of the ND500LK mutant clearly possess crystalloids as evidenced by the characteristic localisation of NTH::GFP in foci associated with pigment clusters (Fig. 3H), and this was observed in all ookinetes examined (n=50). NTH and LAP3 must be present in crystalloids at the same time, because both NTH (this manuscript) and LAP3 (Saeed et al., 2015) are required in their own right for crystalloid biogenesis to happen. Hence, NTH in the absence of LAP3 could not be targeted to crystalloids because the organelle would not form, and *vice versa*. Crystalloids also do not form in the absence of the crystalloid proteins LAP1 [3] or DHHC10 [4] and these collective observations form the basis for the stated hypothesis that different crystalloid proteins interact and pre-assemble into 'crystalloid protein complexes' prior to, and as a prerequisite of, crystalloid biogenesis, as discussed (lines 214-225). We did try to generate double mutants by crossing parasite lines NTH/ND500LK and LAP3/mCherry, but failed to do so in three attempts. This is likely caused by the lower infectivity of the ND500LK sporozoites (Table 1).

The oocyst phenotype of the enzymatic mutant should be quantified as for the structural mutants taking my comments on that quantification in consideration.

As requested, we have now added quantitative data on the oocyst phenotype (i.e. oocyst number and size) of this enzymatically inactive NTH mutant (NTH/ND500LK) in the text and Fig. EV3. This clearly shows that this parasite forms normal oocyst numbers that fail to sporulate and that grow to a larger size, like the other NTH mutants.

- Given the phenotypes of the various functional mutants, I'm surprised that the authors have not provided more data on the localisation of NTH, co-localisation with LAP3 and ACP, and localisation of LAP3 in the functional mutants at in *eg* zygotes and during different time points of oocyst development. I believe this would be a rather important addition to complete the story. We have now provided additional images showing (co)localisation of NTH and LAP3 in zygotes and ookinetes (Fig. 2), which also further support the absence of NTH apicoplast targeting in these life stages. Regarding NTH expression in oocysts, we already mentioned in the manuscript that apart from very young oocysts that still contain an intact crystalloid NTH is not detectable during oocyst development until sporozoites are formed, as shown in Fig. 4.

The single non-sporulated oocyst shown is taken at day 15 and could also represent an unhealthy/dying oocyst, it would help to show earlier time points.

We have replaced the oocyst images in Fig. 3C with those that show more clearly what is going on, and we have artificially coloured the Hoechst DNA stain red to provide better contrast. The images clearly show sporozoites within the NTH/GFP oocysts by brightfield, and this is corroborated by the DNA staining showing the aligned, condensed and elongated sporozoite nuclei. By contrast, the NTH-KO oocyst shows no signs of sporozoite formation in brightfield, whilst DNA staining shows a substantial degree of nuclear expansion indicating that mitosis has occurred to a significant degree. These features are very similar to those of LAP null mutant oocysts that fail to sporulate (e.g. Fig. 7C, [1]). We have also now included additional oocyst images at an earlier point of development (day 11) (Fig. EV3C).

More generally, we did not look at just one oocyst rather at entire oocyst populations across multiple infected mosquitoes before reaching our conclusions regarding sporulation, size and NTH expression. The oocyst images shown in the various figures are representative and were included to illustrate the main phenotypes. We were careful not to select unhealthy/dying oocysts (recognisable by a much more vacuolated appearance).

Additional minor points:

- Please add line numbers for ease of reviewing and commenting

Line numbers have now been added.

- Abstract: only shown for Plasmodium, shouldn't extrapolate to all protozoans

This has been changed as requested.

- Page 3: the apicoplast is not a relic chloroplast but a separately derived plastid of likely red algal origin (Janouskovec et al 2010)

We have now amended the text to reflect this.

- Page 3: The claim of the final sentence is far too broad and not supported by the data, no data are presented for any other than Plasmodium spp and no other organelle than the crystalloid, which to the best of my knowledge has not been identified in any other spp outside the Haemosporina.

We have removed this sentence.

- Page 4: The dispersed NTH-GFP-fluorescence in zygotes (which should be shown, perhaps in the supplement) might be preassembly of crystalloid protein complexes. It would be interesting to confirm this theory of preassembly by investigating the zygote stages / early oocysts of the LAP3-mCherry/NTH-GFP parasite in fluorescence microscopy and check whether they colocalize before assembly in the crystalloids.

We have added an image of the dispersed NTH::GFP fluorescence in zygotes (Fig. 2A), as well as an image showing co-localisation of NTH::GFP and LAP3::mCherry in zygotes (Fig. 2D). We have previously shown using fluorescent protein-tagged LAP3 how this protein is trafficked from the ER to the crystalloid during crystalloid formation in the developing ookinete [1]. We have also shown previously that during this process LAP3 co-localizes with another crystalloid protein DHHC10 [4]. These collective observations are fully consistent with the hypothesis of preassembly of crystalloid protein complexes as discussed in the text (lines 214-225).

- Page 5 & Fig.S1: It would be helpful to indicate in the schematic that the ATS has been removed in NTHΔPP. In general, a little more detail on the genetics would help to clarify how the genome has been manipulated exactly without having to refer to the methods and the primer sequences.

We have now amended this figure (now called expanded view Fig. EV2) and the accompanying M&M section to make it clearer in what way the *nth* allele was modified in each of the mutants.

- Page 7 & Fig.4: The partial co-localisation of NTH and ACP could also be linked to their localisation in membrane vs. matrix.

This is exactly what we were thinking ourselves. We have now amended the text in the manuscript to make this point specifically (lines 262-263).

I'm surprised that the authors chose to generate a gene deletion mutant to integrate the ACP marker rather than to clone the desired promoter (or better still a more constitutive promoter) into the vector

and integrate the vector elsewhere. Better options are available for the stable integration of transgenes. Furthermore, apicoplast and mitochondrion in Plasmodium parasites are closely associated both in intertwined metabolism and physical proximity, with sporozoites forming the exception to the dogma. As I do not think this is common knowledge it would help to refer to Kudryashev et al 2010 to explain this atypical organelle positioning and support the case for NTH targeting to the apicoplast.

We have now added text and the Kudryashev reference to specifically make this point (lines 265-266).

- Page 7: Dual targeting, while not often, is observed for other apicoplast and mitochondrial proteins in Plasmodium. However, given the presented data I find the term dual targeting to be slightly misleading. In no life cycle stage NTH is targeted to two sites (at least based on the data currently presented, see my earlier comment). Stage-specific targeting would be a more suitable term in my opinion. A little speculation on the mechanism of stage-specific targeting would also be welcome. The apicoplast and the ATS are always present, so how could this be overruled to favour the crystalloid assembly/trafficking?

We have changed 'dual targeting' to 'stage-specific targeting' as suggested (line 318). We also now give a possible explanation as to how apicoplast targeting may be overruled in the ookinete (lines 273-275).

- Page 7: In the discussion, the authors completely fail to discuss the discrepancies in apicoplast function (in particular FasII) between *P. berghei* and *P. falciparum* (Van Schaijk et al 2014). Analogous to these differences, NTH could function rather differently in the different Plasmodium spp.

Apologies for the oversight. We now mention in the text that FasII activity in the human malaria parasite *P. falciparum* is required for sporozoite development and that NTH, too, could function differently in *P. falciparum* (lines 308-309).

- A phylogenetic analysis of the various apicomplexan and other NTHs could provide support for the evolutionary history of the single Plasmodium copy.

We share the reviewer's interest in the evolutionary history of this molecule, and in fact we have initiated an extensive phylogenetic analysis of NTH proteins across protozoan, metazoan and bacterial organisms with respect to gene structure, copy number, and structural relatedness.

However, we feel this is not a focus of the current manuscript and will best serve the scientific community when presented in due course as a separate in-depth phylogenetic study, also in view of the limited space available.

- Page 8: There is no support whatsoever for the final statement of the manuscript, nor do the authors suggest in any way what such an intervention strategy should look like. This sentence should be omitted.

As requested, we have now omitted this sentence.

- In the methods there is no section describing how microscopy was performed.

We have added a short section on the microscopy in the M&M section.

Referee #2:

This paper describes a membrane-bound NAD(P) transhydrogenase (NTH) in the malaria parasite. This class of enzymes catalyzes hydride ion transfer between NAD(H) and NADP(H). In metazoans the enzyme is localized in the inner membrane of the mitochondrion. In this study the only malaria parasite NTH is described and found to be localized to the so called crystalloid, a Plasmodium specific organelle, which plays an important role in the development of the parasite in the mosquito. Specifically the crystalloid is needed in the sporogonic oocyst for development of the infectious sporozoites. Previously in a number of studies it has been shown that in the absence of the crystalloid the sporozoites do not develop and transmission is blocked. The authors generated a null mutant of the gene encoding NTH and show that the phenotype of this mutant also affects the development of the sporozoites. They also generated two other mutants in which the phenotype was similar to the null mutant. In the one mutant a truncated form of NTH, lacking an ER signal sequence, was expressed while in the second a point mutation was introduced, which abolished enzymatic activity. The formation of the crystalloid was also investigated in the three mutants and found to be aberrant in the first two mutants while normal in the third. Together these results suggest that NTH is necessary for formation of the crystalloid but its enzymatic activity is not essential for this process. In addition the authors localized the protein to the apicoplast organelle in the sporozoites. The conclusions are solid and supported by the experimental evidence.

This paper presents interesting and important finding as the NTHs in other eukaryotic organisms are mitochondrial enzymes while in the malaria parasite it has an essential role in the parasite-specific organelles crystalloid. It was also detected in the apicoplast, an organelle found in Apicomplexan parasites, though its function in this organelle was not investigated. The results will be interesting to researchers interested in mitochondrial enzymes and working with the mosquito stages of the malaria parasite, possibly also to those studying other Apicomplexan parasites.

I do have some reservations against publication in its present form with the major one being that the function in the sporozoite stage was not investigated. This is certainly not without its challenges and would require at least a few months to a year of work. However, the results would significantly enhance the impact of the paper.

The function of NTH in the sporozoite was already addressed through cross-fertilization of parasite line NTH Δ PP (which does not form crystalloids nor sporozoites) with a parasite that contains a functional *nth* allele (in this case LAP3/mCherry) to bypass the block in sporogenesis in the oocysts and produce NTH Δ PP sporozoites. The fact that these sporozoites are transmissible by mosquito bite, resulting in NTH Δ PP blood stage parasites as shown in Fig. 3E and G, already indicates that functional NTH is not essential for liver stage development of the parasite. To further explore the contribution of NTH to transition from sporozoite to blood-stage parasite, we have carried out and included new experiments investigating the efficacy of NTH/ND500LK sporozoites to cause blood stage infections relative to control (NTH/GFP) sporozoites (lines 278-293, Fig. EV4 and Table 1). Briefly, sporozoites from each population were obtained by crossing the mutants with wildtype (parental) *P. bergeri* parasites. Sporozoites of each GM parasite (recognisable from having a green apicoplast) were then harvested from salivary glands, counted, and known numbers injected intravenously into groups of naive mice. The results obtained show that NTH/ND500LK sporozoites are significantly less effective than their NTH/GFP counterparts in causing intraerythrocytic parasite infection (Table 1), indicating that functional NTH contributes to parasite development in the liver.

The paper is clearly written and put in context with previous literature. However, some figures need improvement. For example a schematic figure of the generated mutants and the crossing strategy (Fig. 2B and 3F) would be very helpful.

Schematic diagrams of the generated mutant alleles are shown in Fig. EV2A. Furthermore, we have now added schematics of the various crosses in Figs. 2C and 3F.

Also the oocyst pictures in Fig. 3C and 3H are not of good quality.

We have replaced the oocyst images in Fig. 3C with those that show more clearly what is going on, and we have artificially coloured the Hoechst DNA stain red to provide better contrast. The images clearly show sporozoites within the NTH/GFP oocysts by brightfield, and this is corroborated by the DNA staining showing the aligned, condensed and elongated sporozoite nuclei. By contrast, the NTH-KO oocyst shows no signs of sporozoite formation in brightfield, whilst DNA staining shows a substantial degree of nuclear expansion indicating that mitosis has occurred to a significant degree. These features are very similar to those of LAP null mutant oocysts that fail to sporulate (e.g. Fig. 7C, [1]).

Figure 3B is problematic. The number of counted midguts should be inserted. The authors should also repeat these experiments at least once.

We have now done an independent experiment counting oocysts of different NTH parasite lines, including mutant NTH/ND500LK (new Fig. EV3). This effectively repeats the experiment shown in Fig. 3B and shows essentially the same thing, namely that NTH is not involved in ookinete-to-oocyst conversion. Information on the number of midguts counted per sample has now been added in the legends. This new figure makes Fig. 3H obsolete and the latter has now been removed.

Referee #3:

This is an interesting, although yet incomplete study, on the sole NTH enzyme (produces NADPH) in *Plasmodium*. If completed, this could also be of interest to scientists outside the malaria research community as it is reported that this enzyme localises to two distinct organelles. It is unfortunately not made clear whether the enzyme plays an essential role in each of these organelles but this can be easily addressed (see my comments below). These organelles are found in two different life cycle forms of the parasite, the ookinete that infects the mosquito and the sporozoite, which is transmitted by the mosquito. In the ookinete NTH localizes to the crystalloid bodies (specific to ookinetes) and in sporozoite to the apicoplast (found in all parasite forms). Through reverse genetic approaches it is shown by gene deletion, domain deletion and point mutation that the protein is essential for crystalloid body and sporozoite formation.

What remains however unclear, is whether the enzyme is important for sporozoite formation due to its role in the crystalloid body or in the apicoplast. This should be experimentally addressed prior to resubmission.

This question has already been addressed: we show in Fig. 3E and G that parasite line NTH Δ PP (which does not form crystalloids or sporozoites) is transmissible by mosquito bite when the block in sporozoite development is bypassed by crossing it with a parasite that contains a functional *nth* allele (in this case LAP3/mCherry). This is because female LAP3/mCherry gametes fertilized by male NTH Δ PP gametes produce ookinetes with normal functional crystalloids, allowing sporogony to proceed normally in the resultant oocyst. Clearly, the successful transmission of NTH Δ PP

sporozoites means that sporozoites lacking apicoplast-resident functional NTH can form. In our view, this shows that the NTH enzyme is important for sporozoite formation due to its role in the crystalloid before sporogony, rather than because of its role in the apicoplast after sporogony. One way such an experiment could be conducted is to cross the existing NTH-GFP line with (1) an NTHdPP-mCherry line and (2) an NTH/ND500LK-mCherry line. This should yield oocysts of mixed color that produce haploid sporozoites. These should be examined for the localization of the NTH and for their infectivity to mice. If the NTH/ND500LK mutation leads to the expected drop of NADPH in the apicoplast, these sporozoites should fail to infect mice and hence the resulting blood stage population should all (or at least a much larger proportion than the input sporozoite population) be NTH-GFP parasites. Similarly, if the NTHdPP fails to localize to the apicoplast, these sporozoites should also fail to infect mice and hence the resulting blood stage population should again all be NTH-GFP parasites. This is an essential experiment to also determine if the enzyme is a potential drug target for malaria prevention.

We thank the reviewer for the suggested experiments. To investigate the contribution of NTH to transition from sporozoite to blood-stage parasite, we have carried out experiments not dissimilar to the ones suggested by this reviewer, but which did not require the generation of new parasite lines thereby reducing the use of laboratory animals (lines 278-293, Fig. EV4 and Table 1). Briefly, NTH/ND500LK sporozoites were obtained by crossing the mutant with wildtype (parental) *P. berghei* parasites. NTH/ND500LK sporozoites (recognisable from having a green apicoplast, Fig. EV4A) were then harvested from salivary glands, counted, and known numbers injected intravenously into groups of naive mice. In parallel, groups of mice were injected with similarly obtained control sporozoites (NTH/GFP). The mice were then subjected to drug treatment to prevent development of wildtype parasites. The results obtained show that NTH/ND500LK sporozoites are significantly less effective than their NTH/GFP counterparts in causing intraerythrocytic parasite infection (Table 1), indicating that functional NTH contributes to parasite development in the liver. We did not do this experiment with NTHdPP, as it has the entire apicoplast transit peptide removed (Fig. EV2) and thus sporozoites are not recognisable from their green apicoplasts.

In addition the paper would benefit from a series of light and electron microscopy images that investigate the difference in sporogony between the NTH-KO and the wild type parasites as well as the localization of the fluorescence of the NTH/ND500LK mutant during sporogony.

As stated in the manuscript, NTH::GFP is not detectable by fluorescence during sporogony until sporozoites are formed, and is found located in the apicoplast (Fig. 4). In the NTH/ND500LK mutant, the NTH protein has been left structurally intact and its apicoplast targeting sequence has not been altered in any way. It is therefore expected to be normally targeted to the sporozoite apicoplast if ND500LK sporozoites were formed. We have confirmed this by looking at sporozoites from a ND500LK x wildtype cross, which shows a subpopulation that display green fluorescence concentrated in a tubular structure (new Fig. EV4A).

Oocyst development of parasite mutants devoid of crystalloids has been a topic of several studies, using both electron and light microscopic cytological assessments [1, 4-7]. The consensus view from this body of work is that such oocyst initially undergo normal growth and mitosis, but ultimately fail to undergo cytokinesis resulting in increased growth and/or degeneration. Our light microscopic observations of NTH-KO oocysts are very similar (e.g. see Fig. 3C) indicating that their developmental progression is similar to that of other crystalloid null mutants reported. In this context, we feel that the added value of EM studies is small relative to the large amount of work that would be required to obtain meaningful data. Instead, we have included additional light microscopic images of oocysts from our parasite lines at the earlier time point of 11 days post-infection. These clearly indicate that up to this time point, at least at cytological level, oocyst differentiation is very similar between the parasite lines examined.

Minor comments:

The paper would also benefit if it would include a life cycle to make it easier for a non-specialist to understand the different forms of the parasite.

We have added a life cycle as requested (Fig. EV1).

Page 3: 13th line of introduction contains a 'they' that should be deleted
A typo. This word has now been deleted.

page 5: lines 5 and 6: please rephrase this statement, the higher molecular weight band could also arise due to an interaction with another protein, hence it is not 'most likely' a homodimer, but 'possibly' a homodimer. This 'could be' consistent with reports...

We have changed 'most likely' to 'possibly' as suggested. Also we have changed 'is fully consistent' to 'is consistent'. In our view 'could be consistent' as proposed as a text change does not read well; things either are consistent or not consistent.

page 8: add a note that at least hints to the possibility that the refractile body in *Eimeria* is somewhat similar to the crystalloid body in *Plasmodium*.

We have now added a sentence stating that the refractile body is postulated to be functionally similar to the Plasmodium crystalloids (lines 326-327).

Explain how the knowledge generated in this paper could lead to novel malaria transmission control strategies.

The discovery that crystalloid function relies on NTH activity indicates that the organelle is dependent on NADPH and involved in biosynthesis. The next phase of this work is to discover exactly what these anabolic processes entail. We may then be able to develop specific inhibitors. On request of reviewer 1, we have omitted the final statement that this work could lead to novel malaria transmission strategies and therefore we will not further speculate on this issue in this manuscript.

Figure 2B, 3E: is there only one crystalloid body in the double fluorescent parasites?

No, see our response to the same comment from reviewer 1. We are now showing cells with two crystalloids.

Figure 1 and 2 could easily be fused to accommodate the necessary new data.

We have added a Table and several Expanded View figures to accommodate the new data.

References

1. Saeed S, Tremp AZ, Dessens JT (2015) Biogenesis of the crystalloid organelle in Plasmodium involves microtubule-dependent vesicle transport and assembly. *Int J Parasitol* **45**: 537-47
2. Stanway RR, Witt T, Zobiak B, Aepfelbacher M, Heussler VT (2009) GFP-targeting allows visualization of the apicoplast throughout the life cycle of live malaria parasites. *Biol Cell* **101**: 415-30, 5 p following 430
3. Carter V, Shimizu S, Arai M, Dessens JT (2008) PbSR is synthesized in macrogametocytes and involved in formation of the malaria crystalloids. *Mol Microbiol* **68**: 1560-9
4. Santos JM, Duarte N, Kehrer J, Ramesar J, Avramut MC, Koster AJ, Dessens JT, Frischknecht F, Chevalley-Maurel S, Janse CJ, *et al.* (2016) Maternally supplied S-acyl-transferase is required for crystalloid organelle formation and transmission of the malaria parasite. *Proc Natl Acad Sci U S A* **113**: 7183-8
5. Raine JD, Ecker A, Mendoza J, Tewari R, Stanway RR, Sinden RE (2007) Female inheritance of malarial lap genes is essential for mosquito transmission. *PLoS Pathog* **3**: e30
6. Saeed S, Lau CI, Tremp AZ, Crompton T, Dessens JT (2019) Dysregulated gene expression in oocysts of Plasmodium berghei LAP mutants. *Mol Biochem Parasitol* **229**: 1-5
7. Saeed S, Tremp AZ, Dessens JT (2018) The Plasmodium LAP complex affects crystalloid biogenesis and oocyst cell division. *Int J Parasitol* **48**: 1073-1078

2nd Editorial Decision

9 December 2019

Thank you for the submission of your revised manuscript to our editorial offices. We have now received the reports from the three referees that were asked to re-evaluate your study, you will find below. As you will see, all three referees have remaining concerns and suggestions to improve the manuscript I ask you to address in a final revised version, either by adding data or text changes, and/or in a detailed point-by-point-response (in case you feel points have already been adequately addressed during the previous revision). Please provide a detailed point-by-point-response in any case.

Further, I have these editorial requests:

- I would suggest a more active title:

NAD(P) transhydrogenase has vital non-mitochondrial functions in malaria parasite transmission

- It seems that the zygote shown in Fig. 2A leftmost panel shows a different orientation than the other two. Could this be rotated?

- Please check that the length all scale bars shown in the images are defined in the respective figure legend.

- Could a scale bar be added to the expanded box in Fig. 4D?

- Finally, please find attached a word file of the manuscript text (provided by our publisher) with

changes we ask you to include in your final manuscript text, and some queries, we ask you to address. Please provide your final manuscript file with track changes, in order that we can see the modifications done.

Thanks for providing the schematic summary figure. I wonder if the content could be expanded a bit, to better summarize the content of the paper, maybe including crystalloid biogenesis and transmission? Please keep the format (jpeg or tiff format with the exact width of 550 pixels and a height of not more than 400 pixels).

REFEREE REPORTS

Referee #1:

The authors have gone to great length addressing most of my concerns and those of the other reviewers. I'm in particularly happy to see they looked in more detail at the infectivity of the NTH/ND500LK sporozoites. I would like to recommend publishing this elegant and interesting study in EMBO reports, even though I still have some reservations. I will leave it to the editors discretion to decide how to deal with these.

I remain puzzled, as I indicated in my initial response, as to why "the authors chose to generate a gene deletion mutant to integrate the ACP marker rather than to clone the desired promoter (or better still a more constitutive promoter) into the vector and integrate the vector elsewhere." In particular, since the deletion of IMC1A has an effect on sporozoite fitness. I still think that "Better options are available for the stable integration of transgenes." So, while the authors' explanation ("We could not use our ACP/mCherry line... ") is convincing they could have simply used a more conventional reference line to also "demonstrate co-localisation or absence thereof in the ookinete and oocyst" as I suggested. At least I think that the authors would do well discussing the limitations of the approach they chose.

The authors state that "We did not cross LAP3/mCherry with the complete knockout (NTH-KO) line, because green fluorescence would reflect the localisation of GFP rather than NTH." However, to proof more convincingly that the crystalloids are not formed a GFP tagged NTH is not required, the random distribution of LAP3-mCherry in double positive as opposed to single coloured parasites, as observed for the other crossing would have been sufficient.

Line 133-134: "indicating that the GFP tag had not interfered with NTH function."
This is only true in combination with the observation that the knockout parasites show a clear phenotype not observed for the tagged parasites. Please, make this clear.

Line 175 and elsewhere: Rather than simply stating n=1000, I think it's better to indicate counted oocyst and midgut numbers from 2 independent experiments.

Referee #2:

The authors have adequately addressed all my comments as well as those of the other reviewers. The changes of the manuscript has much improved. I only have one comment: In the legend to Fig. 3 the panels E and F are switched (line 632 ff).

Referee #3:

The paper has improved by adding an additional experiment that represents an appropriate adaption

of the suggested experiment and the resulting table showing that infection of mice with sporozoites expressing a mutated NTH is delayed compared to a wild type infection.

However, I am amazed the authors just leave it there and don't go the quarter mile to show whether NTH is expressed in liver stages or whether sporozoites expressing mutant versions of NTH are less infectious to cultured liver cells. In my view the paper (although I like it in general but I am also aware that it still is not giving molecular functional insight) can only be published in EMBO with a complete description of the phenotype. Hence the authors need to show if NTH mutant sporozoite migrate normally - this might have to be done with sporozoites that have been salivary gland resident for some time to avoid as much as possible carryover of the wild type NTH protein from the oocyst). The authors should also show through a simple infection assay of cultured liver cells whether NTH mutant sporozoites have an invasion phenotype and whether they have a growth phenotype in the liver. The authors might also choose to do qPCR analysis from livers of infected mice, although I doubt that this is very informative as the presence of wild type parasites might confound the results. Only once these data are presented can their phenotypic analysis be considered complete and can a second function of NTH in transmission be pinpointed.

2nd Revision - authors' response

11 December 2019

Referee #1

The authors have gone to great length addressing most of my concerns and those of the other reviewers. I'm particularly happy to see they looked in more detail at the infectivity of the NTH/ND500LK sporozoites. I would like to recommend publishing this elegant and interesting study in EMBO reports, even though I still have some reservations.

I remain puzzled, as I indicated in my initial response, as to why "the authors chose to generate a gene deletion mutant to integrate the ACP marker rather than to clone the desired promoter (or better still a more constitutive promoter) into the vector and integrate the vector elsewhere." In particular, since the deletion of IMC1A has an effect on sporozoite fitness. I still think that "Better options are available for the stable integration of transgenes." So, while the authors' explanation ("We could not use our ACP/mCherry line...") is convincing they could have simply used a more conventional reference line to also "demonstrate co-localisation or absence thereof in the ookinete and oocyst" as I suggested. At least I think that the authors would do well discussing the limitations of the approach they chose.

Because parasite line NTH/GFP was already drug resistant, we could not introduce a second transgene by sequential transfection and thus opted for the crossing strategy with a ACP::mCherry-expressing line. For the latter, we chose *imc1a* as the target locus, because it was known to be easy to target, and the *imc1a* promoter was known to give good expression levels during sporogenesis. The fact that this would result in a IMC1a knockout phenotype was irrelevant, because the IMC1a knockout forms sporozoites, and because we were interested in heterokaryotic oocysts in which the phenotype is anyway rescued by wildtype IMC1a expressed from the other parental genome. As we outlined in our first response, from our observations of zygotes and ookinetes we were already satisfied that NTH was not present in the zygote/ookinete apicoplast, but with the benefit of hindsight it might have been better to use a different promoter that was also active in these stages to drive ACP::mCherry expression. We now make clear in the main text that ACP::mCherry is driven from a sporozoite-specific promoter (line 264). We also now discuss the limitations of this with regards to the potential NTH localisation in the ookinete apicoplast in the main text (lines 277-279).

The authors state that "We did not cross LAP3/mCherry with the complete knockout (NTH-KO) line, because green fluorescence would reflect the localisation of GFP rather than NTH." However, to prove more convincingly that the crystalloids are not formed a GFP tagged NTH is not required, the random distribution of LAP3-mCherry in double positive as opposed to single coloured parasites, as observed for the other crossing would have been sufficient.

We somewhat missed the point, and the referee is correct. We generated the NTH Δ PP line to see where N-terminally truncated NTH protein would localise (the NTH-KO is no good for this as it expresses only GFP). We then used NTH Δ PP for the LAP3/mCherry cross, because it displayed a NTH knockout phenotype (including absence of crystalloid formation) and also displayed much

better GFP fluorescence levels than the NTH-KO, which made it much easier to detect double positive parasites in ookinete cultures.

*Line 133-134: "indicating that the GFP tag had not interfered with NTH function."
This is only true in combination with the observation that the knockout parasites show a clear phenotype not observed for the tagged parasites. Please, make this clear.
We have now amended the text to make this clear (line 133).*

*Line 175 and elsewhere: Rather than simply stating n=1000, I think it's better to indicate counted oocyst and midgut numbers from 2 independent experiments.
We now state that 1000 oocysts were examined across at least 10 midguts from two independent infections.*

Referee #2

*The authors have adequately addressed all my comments as well as those of the other reviewers.
The changes of the manuscript has much improved. I only have one comment: In the legend to Fig. 3 the panels E and F are switched (line 632 ff).
We thank the reviewer for spotting this oversight. This has now been corrected.*

Referee #3

The paper has improved by adding an additional experiment that represents an appropriate adaption of the suggested experiment and the resulting table showing that infection of mice with sporozoites expressing a mutated NTH is delayed compared to a wild type infection.

However, I am amazed the authors just leave it there and don't go the quarter mile to show whether NTH is expressed in liver stages or whether sporozoites expressing mutant versions of NTH are less infectious to cultured liver cells. In my view the paper (although I like it in general but I am also aware that it still is not giving molecular functional insight) can only be published in EMBO with a complete description of the phenotype. Hence the authors need to show if NTH mutant sporozoite migrate normally - this might have to be done with sporozoites that have been salivary gland resident for some time to avoid as much as possible carryover of the wild type NTH protein from the oocyst). The authors should also show through a simple infection assay of cultured liver cells whether NTH mutant sporozoites have an invasion phenotype and whether they have a growth phenotype in the liver. The authors might also choose to do qPCR analysis from livers of infected mice, although I doubt that this is very informative as the presence of wild type parasites might confound the results. Only once these data are presented can their phenotypic analysis considered complete and can a second function of NTH in transmission pinpointed.

As part of our first revision, we have shown that NTH function is important during the transition from sporozoite to intraerythrocytic parasite, and in doing so we have addressed the reviewers' comments from the first round of review. Whilst we recognise that the research questions addressed by the latest suggested experiments are important, this would involve a second round of major revisions to add mechanistic insight, and we therefore feel these experiments are out of scope of the current study. To avoid problems from using mixed sporozoite populations and potential carry-over of wildtype NTH into mutant sporozoites, we believe these new research questions would be better addressed using an entirely new parasite line expressing a version of NTH that allows sporozoite formation, but is mutated in a way that prevented apicoplast targeting. In our opinion, this is best achieved as part of an in-depth follow-on study.

Corresponding Author Name: Johannes Dessens

Manuscript Number: EMBOR-2019-47832V1